# Tumor Infiltrating Lymphocytes (TILS) and PD-L1 Expression in Breast Cancer: A Review of Current Evidence and Prognostic Implications from Pathologist’s Perspective

**DOI:** 10.3390/cancers15184479

**Published:** 2023-09-08

**Authors:** Giuseppe Angelico, Giuseppe Broggi, Giordana Tinnirello, Lidia Puzzo, Giada Maria Vecchio, Lucia Salvatorelli, Lorenzo Memeo, Angela Santoro, Jessica Farina, Antonino Mulé, Gaetano Magro, Rosario Caltabiano

**Affiliations:** 1Department of Medical, Surgical Sciences and Advanced Technologies G.F. Ingrassia, Anatomic Pathology, University of Catania, 95123 Catania, Italy; giuseppe.broggi@gmail.com (G.B.); giordanatinnirello@yahoo.it (G.T.); lipuzzo@unict.it (L.P.); giadamariavecchio@gmail.com (G.M.V.); lucia.salvatorelli@unict.it (L.S.); jessicafarina2693@gmail.com (J.F.); g.magro@unict.it (G.M.); rosario.caltabiano@unict.it (R.C.); 2Department of Experimental Oncology, Mediterranean Institute of Oncology, 95029 Catania, Italy; lorenzo.memeo@grupposamed.com; 3Pathology Unit, Fondazione Policlinico Universitario Agostino Gemelli IRCCS, 00168 Rome, Italy; angela.santoro@policlinicogemelli.it (A.S.); antonino.mule@policlinicogemelli.it (A.M.)

**Keywords:** breast cancer, TILs, PD-L1, immunotherapy, chemotherapy, triple-negative breast cancer, HER2, luminal breast cancer, CPS

## Abstract

**Simple Summary:**

The aim of our study is to provide a wide perspective on the available literature data on the immune landscape of breast cancers, focusing on TILs and PD-L1 expression across different breast cancer subtypes. Moreover, treatment options such as immunotherapy and chemotherapy in adjuvant and neoadjuvant settings are discussed, along with the most relevant cut-offs and scores for TILs and PD-L1 pathological assessment.

**Abstract:**

With the rise of novel immunotherapies able to stimulate the antitumor immune response, increasing literature concerning the immunogenicity of breast cancer has been published in recent years. Numerous clinical studies have been conducted in order to identify novel biomarkers that could reflect the immunogenicity of BC and predict response to immunotherapy. In this regard, TILs have emerged as an important immunological biomarker related to the antitumor immune response in BC. TILs are more frequently observed in triple-negative breast cancer and HER2+ subtypes, where increased TIL levels have been linked to a better response to neoadjuvant chemotherapy and improved survival. PD-L1 is a type 1 transmembrane protein ligand expressed on T lymphocytes, B lymphocytes, and antigen-presenting cells and is considered a key inhibitory checkpoint involved in cancer immune regulation. PD-L1 immunohistochemical expression in breast cancer is observed in about 10–30% of cases and is extremely variable based on tumor stage and molecular subtypes. Briefly, TNBC shows the highest percentage of PD-L1 positivity, followed by HER2+ tumors. On the other hand, PD-L1 is rarely expressed (0–10% of cases) in hormone-receptor-positive BC. The prognostic role of PD-L1 expression in BC is still controversial since different immunohistochemistry (IHC) clones, cut-off points, and scoring systems have been utilized across published studies. In the present paper, an extensive review of the current knowledge of the immune landscape of BC is provided. TILS and PD-L1 expression across different BC subtypes are discussed, providing a guide for their pathological assessment and reporting.

## 1. Introduction

Breast cancer (BC) exhibits a wide morphological and molecular spectrum of neoplasms with different clinical behavior, prognosis, and response to treatments [1].

On the basis of gene expression patterns, the following molecular subtypes of BC have been identified: (i) luminal-like tumors (the most common subtype of BCs, showing high genetic expression of the estrogen receptor (ER) and lacking significant expression of Erb-B2); (ii) HER2-enriched tumors (characterized by overexpression of Erb-B2 and showing low genetic expression of ER); (iii) basal-like tumors (characterized by expression of genes found in the basal cells of the terminal duct lobular units) [2]. However, the most widely accepted classification of BC is based on the immunohistochemical expression of ER, PR, HER-2, and KI-67, which represent reliable surrogate markers for the molecular classification [2,3]. On this basis, the following subtypes of BC can be identified: (i) luminal A tumors: characterized by high expression of ER and PR, HER2 negativity, and low expression of Ki-67 (less than 20%). This BC subtype shows the best prognosis and a high response to hormone therapy.

(ii) Luminal B tumors: characterized by high expression of ER, low expression/negativity of PR, and high Ki67 (greater than 20%). This subtype shows a worse prognosis compared with Luminal A and may benefit from chemotherapy.

(iii) Triple-negative tumors: characterized by negative immunohistochemical staining for ER, PR, and HER2 and high Ki67. TNBC represents the most aggressive BC subtype, usually presenting in advanced stages and characterized by early relapse and distant metastases. By immunohistochemistry, tumors expressing basal-cell markers such as cytokeratins 5/6, CK17, and CK14 are categorized as basal-like [2,3].

With the rise of novel immunotherapies able to stimulate the antitumor immune response, increasing literature concerning the immunogenicity of BC has been published in recent years [4,5,6,7]. Recent RNA sequencing studies demonstrated the existence of three transcriptome-based subtypes of BC corresponding to different immune categories: immune high, medium, and low [8].

Immune-high tumors are characterized by the highest expression of tumor-infiltrating lymphocytes (TILS) as well as PDL1 [8]. Triple-negative breast cancer (TNBC) and HER2+ tumors are frequently included in the immune-high category and represent potential responders to immunotherapies [8]. On the other hand, immune medium and immune low tumors show little to no immune cell infiltration and are unresponsive to immunotherapies [8]. Interestingly, estrogen receptor (ER) and progesterone receptor (PR)-positive tumors usually fall into the immune medium and low groups [9]. In recent years, numerous clinical studies have been conducted in order to identify novel biomarkers that could reflect the immunogenicity of BC and predict response to immunotherapy [10]. In this regard, TILs have emerged as an important immunological biomarker related to the antitumor immune response in BC [4,5]. TILs are more frequently observed in triple-negative breast cancer and HER2+ subtypes, where increased TIL levels have been linked to a better response to neoadjuvant chemotherapy and improved survival [4,5]. PD-L1 is a type 1 transmembrane protein ligand expressed on T lymphocytes, B lymphocytes, and antigen-presenting cells and is considered a key inhibitory checkpoint involved in cancer immune regulation [6]. The PD-1/PD-L1 pathway plays a crucial immunoregulatory role by suppressing the immune system in both physiological and pathological conditions, including cancer. PD-L1 ligand specifically binds to the PD-1 receptor expressed on T-lymphocytes, leading to a decrease in the immune response [6,7,8,9,10].

PD-L1 ligand is expressed by several inflammatory cells, including activated T-lymphocytes, B-lymphocytes, macrophages, and dendritic cells [6,7,8,9,10]. Moreover, tumor cells can also express PD-L1 as an “adaptive immune mechanism” to remain undetected by the immune system [6,7,8,9,10]. Additionally, PD-L1 exerts pro-tumorigenic activity by promoting tumor growth and survival [6,7,8,9,10].

PD-L1 immunohistochemical expression in breast cancer is observed in about 10–30% of cases and is extremely variable based on tumor stage and molecular subtypes [11]. Briefly, TNBC shows the highest percentage of PD-L1 positivity, followed by HER2+ tumors [11]. On the other hand, PD-L1 is rarely expressed in hormone-receptor-positive BC [11,12].

The prognostic role of PD-L1 expression in BC is still controversial since different immunohistochemistry (IHC) clones, cut-off points, and scoring systems have been utilized across published studies [12,13].

The prognostic role of PD-L1 expression in BC was demonstrated for the first time by Muenst et al. in 2015 [14]. In this study, positive staining (both membranous and cytoplasmic) for PD-L1 was observed in 152 out of 650 BC patients, and a significant correlation was observed between PDL-1 positivity and several clinicopathological parameters (large tumor size, lymph node involvement, tumor grade, ER negativity, HER2-positive tumors, and high Ki67 index) [14].

Despite this association with worse clinico-pathologic features, several studies highlighted that PD-L1 expression could predict a better response to chemotherapy and a better prognosis, mainly in the TNBC subtype [11,13]. Moreover, a significant correlation between PD-L1 positivity and TIL scores has been documented [11,13].

In the present paper, an extensive review of the current knowledge of the immune landscape of BC is provided. Moreover, TILS and PD-L1 expression across different BC subtypes are discussed, providing a guide for their pathological assessment and reporting.

## 2. Tumor-Infiltrating Lymphocytes in Breast Cancer

Increasing scientific evidence suggests the prognostic and predictive role of TILs in breast cancer [15,16]. In this regard, the presence of TILs within a tumor is strictly related to the anti-tumor host immune response [15,16].

TILs are constituted by all mononuclear cells (lymphocytes, plasma cells, monocytes, and NK-T cells) dispersed in the tumor stroma (stromal TILs) or located within the tumor (intratumoral TILs) [15,16,17].

Based on their phenotype, TILs can be classified as CD8+ T cells, CD8+ tissue-resident memory T cells, CD4+ T helper cells, CD4+ regulatory T cells, CD4+ follicular helper T cells, and tumor-infiltrating B cells. However, the specific role and clinical significance of each TIL subpopulation are still uncertain [15,16,17,18].

According to the recommendations for assessment of TILs in breast cancer proposed by the “International Working Group for TILs in Breast Cancer”, the pathological assessment of TILs should include only stromal TILs [19]. TILs evaluation is performed on hematoxylin and eosin (H&E)-stained sections by evaluating the ratio between the intratumoral stromal area containing lymphocytes and plasma cells and the total intratumoral stromal area [19]. According to the percentages of stromal TILs, three different groups can be identified: low TILs (0–10% immune cells in stromal tissue within the tumor), intermediate TILs (11–40%), and high TILs (>40%) (Figure 1). [19]

Following the above-mentioned recommendations, the prognostic and predictive role of TILs has been investigated by several studies and clinical trials, mainly in triple-negative and HER-2-positive breast cancer patients (Table 1) [17,18,20].

### 2.1. TILs in Triple Negative Breast Cancer

TNBC is a breast cancer subtype lacking expression of ERs, HER2, and PRs and characterized by a poor prognosis [21]. TNBC shows higher levels of TILs, a higher tumor mutation burden (TMB), and high PD-L1 expression [21]. Increasing literature data indicate that high levels of TILs in TNBC are significantly related to a better response to chemotherapy as well as a better prognosis [22]. TNBC represents the most immunogenic BC subtype [8]. The high immune cell infiltrate observed in these tumors is thought to represent an adaptive mechanism of the immune system to prevent tumor growth and metastatic spread [8,22]. Therefore, in TNBC patients, an increase in TILs levels, which represent the anti-tumor host immune response against cancer cells, has been related to improved response to chemotherapy and longer survival [8,22]. Moreover, chemotherapy treatments may also boost the antitumor immune response [8,22].

Concerning the prognostic role of TILs in early TNBC, two studies, including 2148 patients treated with adjuvant chemotherapy and 906 women treated with neoadjuvant chemotherapy, respectively, demonstrated the clinical utility of TIL evaluation [4,23].

Briefly, high TILs have been demonstrated to predict responses to adjuvant and neoadjuvant chemotherapy with anthracycline; moreover, each 10% increment in TILs was significantly related to longer disease-free survival and overall survival [4,23]. On the other hand, low-TILs are more frequently detected in patients with older ages, larger tumor sizes, and lymph node metastases [4,23].

Based on these findings, the World Health Organization (WHO) classification of tumors (5th edition) strongly suggests TIL assessment in TNBC and HER2+ subtypes as a prognostic biomarker [24].

Several scientific studies concerning the predictive role of TILs for immune therapy or combined immune therapy/chemotherapy in TNBC are emerging [19,21]. Results of previous studies in early-stage TNBC suggest that high TILs predict response to neoadjuvant immune therapy alone or in combination with chemotherapy [19,21].

Similar predictive roles of TILs for immune checkpoint inhibitor therapy have also emerged in advanced/metastatic TNBC [25].

### 2.2. TILs in HER2+ Breast Cancer

HER2-positive breast cancer accounts for approximately 15–20% of all breast carcinomas and is considered a biologically aggressive subtype [26,27]. Due to their low mutational burden, HER2-positive BCs are generally considered “cold” tumors [27]. However, recent studies have started to explore immunotherapeutic approaches to target HER2-positive tumors both in neoadjuvant and adjuvant settings [20,28,29]. Regarding the latter, the results of the FIN-HER study and ShortHER trial showed that high TIL levels were related to longer overall survival and an improved response to trastuzumab compared with low TILs [20,29]. However, in contrast to these data, Perez et al. reported that breast tumors with high TILs showed a worse response when combining trastuzumab with chemotherapy than those treated with chemotherapy alone [30]. Moreover, the analysis from the N9831 trial demonstrated that high TILs predicted a lack of response to trastuzumab [31]. Therefore, given the conflicting results concerning the role of TILs in predicting response to adjuvant chemotherapy, further studies on larger cohorts are still needed to understand TILs biological role in HER-2 breast tumors.

TILs evaluation in a post-NAD setting has also been associated with breast cancer prognosis [5,32,33]. According to the results of three recent meta-analyses, high TILs are significantly related to a better response to neoadjuvant chemotherapy plus trastuzumab, regardless of the type of neoadjuvant regimen [22,30,31]. Moreover, a statistically significant correlation between high TILs and improved prognosis has also emerged [5,23,32].

Concerning advanced/metastatic HER-2-positive breast tumors, the prognostic and predictive role of TILs is still controversial. According to a retrospective analysis of the patients enrolled in the CLEOPATRA trial, high TILs were associated with increased OS; on the contrary, TIL count showed no significant prognostic or predictive value in the analysis of the MA.31 phase 3 trial [33,34].

Lastly, controversial data from the metastatic setting have also emerged when evaluating the prognostic and predictive role of TILs in patients receiving immunotherapy [35,36]. Therefore, further studies are needed to investigate the interactions between the immune system and tumor cells in HER2+ breast cancer.

### 2.3. TILs in Hormone-Receptor+/HER2− Breast Cancer

The prognostic and predictive role of TILs in hormone-receptor-positive (HR+) and HER-2-negative breast cancer subtypes (luminal A and luminal B tumors) is still poorly established.

In this regard, HR+/HER2− BC is associated with a low TIL count and a lower TMB [13]. Moreover, ER expression has been related to decreased MHC class II expression on lymphocytes, suppression of interferon-γ (IFN-γ) signaling, and decreased activity of CD8+T-cells [15,37]. However, given the wide morphological and biological heterogeneity of luminal A and luminal B tumors, several studies have tried to identify “immunogenic” subgroups. Briefly, it has been shown that TIL positivity is more frequently detected in luminal B subtypes [15,22].

Concerning the predictive and prognostic role of TILs in early HR+/HER2− BC, conflicting results have been highlighted: a significant association between high TILs and a worse prognosis has emerged in some studies, while other authors failed to demonstrate the prognostic significance of TILs [22,37]. Moreover, TILs have been associated with a poor response to aromatase inhibitor therapy in HR+/HER2− BC [22,37].

Regarding TIL subpopulations, limited literature data are still available; however, a recent retrospective analysis of 563 patients with early HR+/HER2− BC documented that high CD8+ sTILs were more frequently detected in patients with *PIK3CA*-mutated tumors and that they were related to a higher risk of recurrence [38].

Based on these preliminary findings, routine assessment of TILs in HR+/HER2− BC is still not recommended and cannot be considered a prognostic or predictive biomarker. 

**Table 1 cancers-15-04479-t001:** Prognostic and predictive roles of TILs in different breast cancer subtypes.

BC Subtypes	TILS Levels	Prognostic Role	Predictive Role	Pathological Assessment	References
TNBC	High (>40%)	Yes	Yes to adjuvant and neoadjuvant chemotherapy	Recommended	Bianchini, 2021 [21] Stanton, 2016 [22] Denkert, 2018 [4]
HER2+	High (>40%)	Yes	Yes to neoadjuvant chemotherapy + immunotherapy	Recommended	Nuciforo, 2017 [28] Loi, 2014 [20] Dieci, 2019 [29]
HR+	Low/intermediate (0–10%/11–40%)	Yes	Not fully established	Not recommended	El Bairi, 2021 [15] Stanton, 2016 [22] Valenza, 2023 [37]

Abbreviations: BC—breast cancer; TNBC—triple-negative breast cancer; HR+—hormone receptor positive breast cancer.

## 3. PD-L1 Pathway, General Considerations

PD-L1 is a type 1 transmembrane protein ligand generally expressed on T lymphocytes, B lymphocytes, and antigen-presenting cells; it is involved in the immune regulation of several physiological and pathological conditions, including pregnancy, antigen presentation to T lymphocytes, tissue and organ transplants, infectious diseases, and cancer [6].

Following the binding of PD-L1 with the PD-1 or B7.1 (CD80) receptors, a suppressive signal is transmitted to T lymphocytes, leading to a decrease in the immune response (Figure 2) [6,11,12].

Moreover, the intracellular signals transmitted by PD-L1 promote neoplastic cell proliferation and inhibit pro-apoptotic signals mediated by interferons (Figure 2) [6,11,12].

In the last few years, several immunotherapeutic molecules capable of inhibiting the PD-1/PD-L1 axis have been shown to improve the immunological response by inducing T cells to recognize and suppress cancer cells [13].

Based on these findings, several monoclonal antibodies targeting the PD-1 receptor and PD-L1, so-called immune checkpoint inhibitors, have been successfully utilized in clinical practice; these include Pembrolizumab and Nivolumab (targeting the PD-1 receptor), Atezolizumab, Avelumab, and Durvalumab (inhibiting the PD-L1 ligand) [13].

In clinical practice, the immunohistochemical evaluation of PD-L1 expression in neoplastic tissues is the gold standard method for selecting patients eligible for immune checkpoint inhibitor therapy. In this regard, several immunohistochemical assays have been developed for PD-L1 evaluation. The most common PD-L1 assays used in clinical trials include SP142, 28-8, 22C3, and SP263, each of which has been validated with specific platforms [12,13,39].

Accordingly, different immunohistochemical scoring systems have been proposed for quantifying PD-L1 expression in different neoplastic tissues: (i) tumor proportion score (TPS), which evaluates the percentage of positive PD-L1 neoplastic cells among all viable tumor cells; (ii) combined proportion score (CPS), namely the ratio between all PD-L1 positive neoplastic cells and inflammatory cells and the total number of viable tumor cells multiplied by 100; (iii) immune cell score (IC), which takes into account the percentage of the area occupied by PD-L1-positive immune cells relative to the whole tumor area (Table 2) [12,13,39].

Tumor-infiltrating lymphocytes, recruited by tumor antigens, release cytokines, including IFN-γ, which increases the expression of PD-L1. Following the binding of PD-L1 to the PD-1 receptor, a suppressive signal is transmitted to T-cells; moreover, an anti-apoptotic signal is transmitted to tumor cells, leading to tumor survival and T-cell dysfunction.

### Temporal and Spatial Heterogeneity of TILs and PD-L1 Expression during Metastatic Progression

Recent studies have demonstrated extensive discrepancies in TIL count and PD-L1 expression among primary tumors and their paired metastases [44]. Biomarker analyses based on the clinical trial IMPassion130 documented a highly heterogeneous PD-L1 expression between primary tumors and their paired metastases [42]. Higher PD-L1 concordance has been documented in synchronous tumor samples compared with metachronous ones; additionally, a greater clinical benefit with immunotherapy was observed when PD-L1 status was evaluated in metastatic sites [11,45].

The temporal evolution of TILS has been reported by several studies, which documented lower TIL counts at metastatic biopsies compared with primary tumors [11,45].

A recent meta-analysis on this topic demonstrated a significant decrease in PD-L1 expression at metastatic sites [45]. This reported discordance of PD-L1 expression was bi-directional since 50% of patients with PD-L1 positive primary tumors showed PD-L1 negative metastatic biopsies; conversely, about 30% of patients with PD-L1 negative primary tumors were PD-L1 positive in their metastatic biopsies [45].

However, to date, little is known about the optimal metastatic site for PD-L1 evaluation since the few available studies demonstrated a significant difference in PD-L1 expression between different metastatic sites from the same patient [11,42,45]. Further studies on this topic are needed to clarify this reported discordance; however, based on the current literature evidence, PD-L1 assessment in metastatic bioptic samples should be assessed when possible to achieve a more accurate treatment strategy.

## 4. PD-L1 and Immunotherapy in Breast Cancer Subtypes

Immune checkpoint inhibitors (ICI) have recently emerged as a novel immunotherapeutic approach capable of improving the anti-cancer immune response by targeting specific immunologic receptors on the surface of T-lymphocytes [6,7]. These latter receptors are immunological checkpoints composed of inhibitory and stimulatory pathways that maintain a balance between pro-inflammatory and anti-inflammatory signals [6,7,10]. Therefore, ICIs have been specifically developed to target the most relevant immune inhibitory receptors, including CTLA-4, PD-1, and PD-L1 [6,7,10]. Other antibodies are still in clinical development to target additional immune checkpoints such as B7H3, CD39, CD73, the adenosine A2A receptor, CD47, LAG-3, and TIM-3 [6,7,10].

To date, the major clinical benefits of ICIs in breast cancer are restricted to the inhibition of the PD1/PD-L1 pathway [6,7,10]. The combination of PD1/PD-L1 blockade with CTLA-4 inhibition has been investigated in a single-arm pilot study where a response rate of 43% was observed in patients with metastatic TNBC; on the other hand, patients with HR+ breast cancer showed no significant responses [46]. With the increasing use of ICIs in BC treatment, several adverse events related to the enhanced immune response induced by these therapies [47]. Adverse events related to ICIs are the result of an autoimmune response that can affect any organ of the body [47]. The most common manifestations include dermatitis, diarrhea, colitis, and endocrine dysfunctions mainly affecting the thyroid, hypophysis, and adrenal glands [47]. Pneumonitis and myocarditis represent rare but potentially fatal adverse events [47]. Moreover, combination therapies (ICIs plus chemotherapy; dual immunotherapy) are associated with a higher incidence of adverse events [47].

### 4.1. Hormone-Receptor Positive/HER2 Negative Breast Cancer

In the HR-positive and HER2-negative BC subgroups, PD-L1 immunoreactivity is documented in up to 9% of luminal A and 42% of luminal B subtypes [48]. A significant decrease in PD-L1 expression is observed in metastatic tumors, among which 0–1% and 10–12% positive rates are reported in luminal A and luminal B patients, respectively [48]. Given the wide heterogeneity of PD-L1 and TILs in this BC subgroup, only a few studies have considered an immunotherapeutic approach in patients with PD-L1+/ER+/HER2− BC [49]. Promising data have been highlighted by the KEYNOTE-028 and I-SPY2 trials, in which an improved pathological complete response was observed in PD-L1-positive tumors [49,50]. However, results are still preliminary, and significant clinical benefits of immunotherapy have not been reported yet.

Despite the fact that invasive lobular breast carcinoma (ILC) represents the second most common BC subtype, only a few data points are currently available on the immune landscape of this tumor. Interestingly, in the KEYNOTE-028 trial, two patients with PD-L1-positive and ER-positive BC showed a pathological response to pembrolizumab [51].

Moreover, studies based on transcriptomic profiling demonstrated the existence of an immune-related ILC subtype characterized by a high immune infiltrate and the expression of immune-related genes [52]. In this regard, the GELATO trial represents the first clinical trial to investigate the benefits of immune checkpoint inhibitors in patients with ILC [53]. In this study, four partial responses and two stable diseases were observed in patients with metastatic ILC treated with carboplatin and atezolizumab [53]. However, most responders (4/6 patients) had triple-negative ILC; therefore, the antitumor activity of immune checkpoint inhibitors, especially in patients with HR+ ILC, is still a matter of debate [53]. Additional trials specifically designed for ILC patients are needed to establish the role of immunotherapy in this BC subtype.

### 4.2. HER2 Positive Breast Cancer

In early-stage HER2+ BC tumors, PD-L1 immunoreactivity is documented in around 30% of cases, whereas a significant decrease in PD-L1 expression (9–10%) is observed in metastatic tumors [54,55].

However, the prognostic significance of PD-L1 in HER2+ BC is still controversial since some studies documented poor outcomes in metastatic HER2+/PD-L1+tumors, whereas other authors documented improved survival in patients with high levels of PD-1/PD-L1 expression [13,55].

Some studies are currently investigating the clinical benefits of immunotherapy with anti-PD-1/PD-L1 antibodies combined with trastuzumab. In this regard, preliminary results from the phase II randomized KATE2 trial indicate an improved OS in PD-L1-positive patients affected by locally advanced or metastatic BC treated with the combination of atezolizumab and ado-trastuzumab emtansine (T-DM1) [56]. Moreover, the phase Ib/II PANACEA trial documented a 15% response rate in HER2+ advanced BC patients treated with the combination of pembrolizumab and trastuzumab [36].

### 4.3. Triple-Negative Breast Cancer

TNBC encompasses a wide morphological and molecular spectrum of neoplasms, as recent RNA sequencing studies demonstrated the existence of several transcriptome-based subtypes of TNBC (luminal androgen receptor, immunomodulatory, basal-like immune-suppressed, and mesenchymal-like) [8]. In this regard, only the immunomodulatory subtype would benefit from immunotherapy.

In this scenario, PD-L1 expression in tumor and immune cells, evaluated by immunohistochemistry, has become a crucial step for selecting potential responders to immunotherapy (Table 3) [12].

In early-stage TNBC tumors, PD-L1 immunoreactivity is documented in around 45 to 55% of cases; metastatic patients show higher PD-L1 staining percentages (around 35% of cases) compared with other BC subtypes [12]. Moreover, a significant increase in PD-L1 expression following immunotherapy has been documented in the PCD4989g trial [57].

As far as prognosis is concerned, several studies demonstrated better overall survival in PD-L1-positive patients; moreover, *BRCA1* gene mutations and expression of cytotoxic T-lymphocyte antigen 4 (CTLA-4) are more frequently detected in PD-L1+ TNBC [58].

## 5. PD-L1 Assays and Immunohistochemical Scores: Results from Clinical Trials

Several PD-L1 IHC assays are currently approved for predicting the response to anti-PD1 and anti-PD-L1 immunotherapy [12,13,39]. Validated PD-L1 clones for TNBC include SP142, 28-8, SP263, and 22C3, each linked to different therapies [12,13,39].

In TNBC, clinical trials proposed several PD-L1 IHC clones, staining platforms, scoring systems, and cut-offs [12,13,39]. Based on the results of the Keynote-355 trial, the PD-1 inhibitor Pembrolizumab, in combination with chemotherapy, has been approved by the FDA for patients with unresectable/metastatic TNBC showing a PD-L1 CPS ≥ 10 [40]. Following the Keynote-522 trial results, neoadjuvant pembrolizumab in combination with chemotherapy has been approved for high-risk early-stage TNBC as treatment; following surgery, monotherapy with pembrolizumab is approved as adjuvant therapy [41]. In the Keynote-522 trial, tumor samples were considered PD-L1-positive based on a CPS ≥ 1; however, a better pathological complete response in patients treated with pembrolizumab was observed regardless of the PD-L1 immunohistochemical expression [41].

Based on the results of the IMpassion130 trial, the Ventana PD-L1 SP142 test (IC score ≥ 1%) has been approved by the FDA for the selection of advanced TNBC or mTNBC suitable for combination therapy with Atezolizumab plus nab-paclitaxel [42]. Based on the same cohort, a post hoc harmonization study attempted to evaluate the inter-assay variability of three PD-L1 assays, namely the Ventana SP142 and SP263 and the Dako 22C3 [59]. IC score (≥1%) was utilized for SP142 and SP263 assays, while CPS score (≥1) was evaluated for the 22C3 assay [59]. Interestingly, 22C3 and SP263 clones individuated a higher rate of PD-L1 positive patients (81% and 75%, respectively) compared with the SP142 clone (46%) [59]. Despite the lower number of PD-L1-positive tumors detected, patients that were only positive for the SP-142 assay showed higher progression-free survival (4.2 months) compared with patients that were only positive for the 22C3 assay or SP-263 clones [59].

In early TNBC, the Impassion031 trial demonstrated that the clinical benefits of neoadjuvant atezolizumab were independent from the PD-L1 IC (SP142) status [43].

### Relevant PD-L1 Scoring Methods

In TNBC, IC and CPS have been specifically validated for the selection of patients eligible for atezolizumab and pembrolizumab therapies, respectively (Table 4) [12,13,39].

Immune Cell Score.

The IC is specifically developed for the VENTANA PD-L1 (SP142) assay. It takes into account all PD-L1-positive immune cells (lymphocytes, macrophages, granulocytes, dendritic cells, and plasma cells) located intratumorally or in a small peritumoral stromal rim [12,13,39]. Briefly, immune cells are scored as the percentage of the area occupied by all PD-L1-positive immune cells relative to the whole tumor area [12,13,39]. A PD-L1 IC score of ≥1% is considered adequate for atezolizumab therapy [12,13,39].

Combined Positive Score.

CPS is specifically developed for the 22C3 (pharmDx) assay [12,13,39]. CPS is the number of PD-L1-positive tumor cells and intratumoral immune cells divided by the total number of viable tumor cells, multiplied by 100 [12,13,39]. PD-L1 CPS score ≥ 10 is considered adequate for pembrolizumab combination therapy in inoperable/metastatic TNBC patients [12,13,39].

## 6. Limitations of PD-L1 IHC

To date, the main limitations of PD-L1 IHC testing are related to the presence of several immunohistochemical assays for PD-L1 evaluation paired with different immunohistochemical scoring systems. Despite several harmonization studies that attempted to evaluate the inter-assay variability of different PD-L1 assays, the presence of several PD-L1 IHC clones, staining platforms, scoring systems, and cut-offs has generated confusion among pathologists and oncologists. Moreover, the temporal and spatial heterogeneity of TILs and PD-L1 expression among primary tumors and their paired metastases may also account for the suboptimal response rates to PD-L1 inhibitors.

Therefore, future research should focus on the interchangeability of the IHC assays in order to provide definitive guidelines for PD-L1 assessment in breast cancer. The optimal anatomical site for PD-L1 testing in metastatic tumors should also be clarified.

## 7. Conclusions

Although immune checkpoint inhibitors have undoubtedly represented a major step forward in the therapy of many malignancies, including breast cancer, it is widely accepted that not all patients may undergo immunotherapy; accordingly, the accurate selection of breast tumors that may benefit from these relatively novel therapeutic approaches is one of the most debated topics of BC oncological research [60,61,62]. In this regard, the study of TILs and the immunohistochemical evaluation of PD-L1 immunoreactivity among different BC subtypes are actually considered the most reliable markers capable of predicting response to immunotherapy. Although it has been shown that PD-L1-positive BCs frequently exhibit high levels of TILs and a lack of expression of ER, PR, and HER2, and that patients affected by TNBC are the ones who may benefit the most from immunotherapy, promising results have been obtained in some trials also for patients with HR+ and/or HER2+ BC [63]. Accordingly, further studies are needed to validate these preliminary data, but it seems likely that in the future, the use of immunotherapy could also be extended to the other subtypes of BC.

This review also highlights that the predictive and prognostic role of TILs in BC is still controversial and under investigation. In this regard, despite the correlation between TILs and better prognosis/response to chemotherapy in TNBC, a worse outcome has been documented in other BC subtypes exhibiting high TILS. Some studies indicated a worse prognosis and poor response to aromatase inhibitors in HR+/HER2− BC; moreover, a worse response to trastuzumab has been documented in HER2+ tumors exhibiting high TILs.

## Figures and Tables

**Figure 1 cancers-15-04479-f001:**
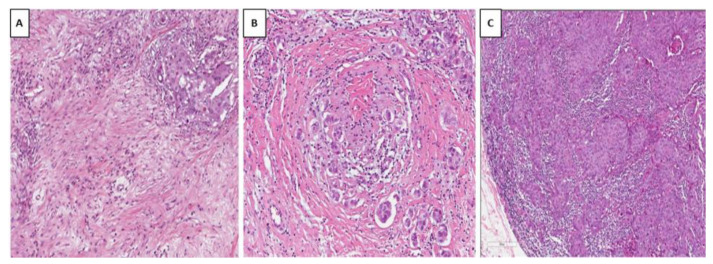
Hematoxylin and eosin (H&E)-stained sections illustrating breast carcinomas with different TIL distributions in our series. (**A**) Invasive breast carcinoma with low TILs (0–10%) in stromal (H&E, ×20). (**B**) Another case showing increased TILs in tumor stroma, categorized as intermediate (11–40%) (H&E, ×20). (**C**) Invasive carcinoma of the breast showing high TILs (>40%) in tumor stroma (H&E, ×20).

**Figure 2 cancers-15-04479-f002:**
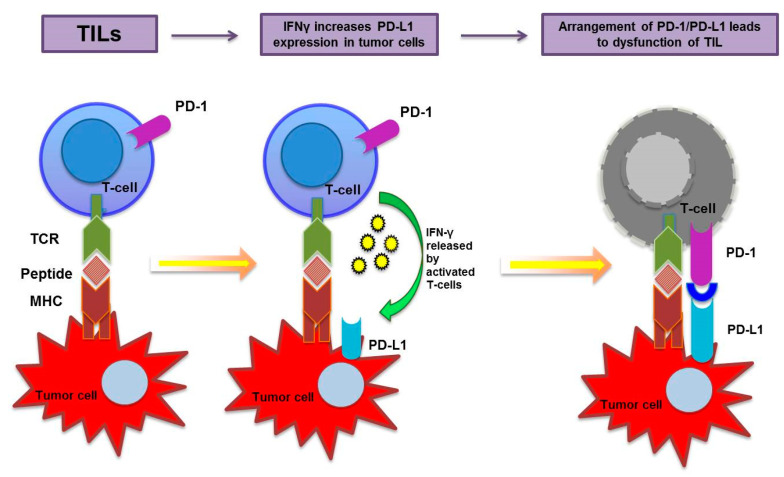
PD-L1-mediated immune escape.

**Table 2 cancers-15-04479-t002:** FDA-approved PD-L1 assays in clinical practice.

ICI	PD-L1 Assay	PD-L1 Score	Setting	Therapy	References
Pembrolizumab	22C3 (pharmDx)	CPS ≥ 10	Unresectable/metastatic TNBC	Pembrolizumab plus chemotherapy	Cortes, 2022 [40]
		CPS ≥ 1/regardless of PDL1 status	high-risk early-stage (NAD/AD)	Pembrolizumab plus chemotherapy as neoadjuvant treatment, and then continued as a single agent as adjuvant therapy	Downs-Canner, 2023 [41]
Atezolizumab	SP142 (Ventana)	IC score ≥ 1	Unresectable/metastatic TNBC	Atezolizumab plus nab-paclitaxel	Emens, 2021 [42]
		Regardless of IC	NAD	Atezolizumab	Mittendorf, 2020 [43]

Abbreviations: ICI—immune checkpoint inhibitor; TNBC—triple-negative breast cancer; NAD—neoadjuvant chemotherapy; AD—adjuvant chemotherapy; CPS—combined proportion score; IC—immune cell score.

**Table 3 cancers-15-04479-t003:** Therapeutic approach in advanced TNBC based on PD-L1 IHC.

PD-L1 Status	FDA-Approved PD-L1 Scores	Therapy	Relevant Clinical Trials
PD-L1-negative	CPS < 10 (22C3) IC < 1 (SP142)	No Immunotherapy	
PD-L1-positive	CPS < 10 IC score ≥ 1%	Atezolizumab plus Nab-paclitaxel	IMpassion130 [42]
PD-L1-positive	CPS ≥ 10 IC score ≥ 1%	Pembrolizumab/Atezolizumab plus chemotherapy (Nab.paclitaxel or Carbo/Gem or paclitaxel)	Keynote-355 [40] Keynote-522 [41] IMpassion130 [42]
PD-L1-positive	CPS ≥ 10 IC score < 1%	Pembrolizumab plus chemotherapy (Nab.paclitaxel or Carbo/Gem or paclitaxel)	Keynote-355 [40] Keynote-522 [41]

Abbreviations: CPS—combined proportion score; IC—immune cell score.

**Table 4 cancers-15-04479-t004:** PD-L1 scoring methods.

PD-L1 Scoring Methods	Assay	Scoring Method	ICI
IC	Ventana (SP142)	All PD-L1-positive immune cells located intratumorally or in a small peritumoral stromal rim	Atezolizumab
CPS	DAKO (22C3)	The number of PD-L1 positive tumor cells and intratumoral immune cells divided by the total number of viable tumor cells, multiplied by 100	Pembrolizumab

## Data Availability

The data can be shared up on request.

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
