# Peer review of "Tumor Infiltrating Lymphocytes (TILS) and PD-L1 Expression in Breast Cancer: A Review of Current Evidence and Prognostic Implications from Pathologist’s Perspective"

_cancers, 2023, doi:10.3390/cancers15184479_

Round 1

Reviewer 1 Report

The manuscript examines the utilization of PD-L1 immunotherapy in breast cancer treatment, highlighting the correlation between PD-L1 positivity, high tumor-infiltrating lymphocyte levels, and the absence of ER, PR, and HER2 expression. Notably, triple-negative breast cancer (TNBC) patients have demonstrated the most favorable response to immunotherapy. Accurate patient selection for immunotherapy is crucial, with the evaluation of the tumor immune microenvironment and PD-L1 immunoreactivity through immunohistochemistry being reliable predictive markers. While TNBC patients are the primary candidates, promising outcomes have been observed in hormone receptor-positive (HR+) and/or HER2-positive (HER2+) breast cancer patients in certain trials. Further studies are needed to validate these findings and potentially extend immunotherapy to other breast cancer subtypes.

To enhance the manuscript, it would be beneficial for the author to delve into the mechanism of action of immune checkpoint inhibitors and their interaction with the immune system. This section could provide insight into the scientific basis and potential of immunotherapy in breast cancer treatment. Additionally, a comprehensive analysis of the potential side effects and limitations of immunotherapy should be included to present a balanced perspective. Exploring other emerging immunotherapeutic approaches (not just PD-L1 therapy) or combination therapies for breast cancer treatment would further enrich the manuscript and provide a broader overview of the current research landscape.

Author Response

We thank the reviewer for its thoughtful comments and suggestions. We have incorporated changes that reflect the detailed suggestions you have graciously provided. To facilitate your review of our revisions, the following is a point-by-point response to the questions and comments. All changes in the main text have been highlighted in yellow.

Q1: To enhance the manuscript, it would be beneficial for the author to delve into the mechanism of action of immune checkpoint inhibitors and their interaction with the immune system. This section could provide insight into the scientific basis and potential of immunotherapy in breast cancer treatment.

Q2: Additionally, a comprehensive analysis of the potential side effects and limitations of immunotherapy should be included to present a balanced perspective.

Q3: Exploring other emerging immunotherapeutic approaches (not just PD-L1 therapy) or combination therapies for breast cancer treatment would further enrich the manuscript and provide a broadera overview of the current research landscape.

A1-A2-A3: According to reviewer’s suggestion, section 3.2 has been implemented with additional insights about Immune Checkpoint Inhibitors and their interaction with the immune system. Additionally, other potential immunotherapeutic targets as well as adverse events related to immunotherapy have been discussed.

Reviewer 2 Report

This is a well written and easy to read review on the current state of play of TIL and PD-L1 quantification in breast cancer. 

The tables should be improved by adding references and a legend detailing whether/which regulatory body has made those recommendations (e.g. FDA).

The authors should discuss why the breast cancer subtype with the worst outcome (TNBC) has the highest number of TILs and yet they are prognostic and predictive of better response to chemotherapy. Furthermore, in some of the other breast cancer subtypes high TILs have been shown in some studies to be associated with a worse prognosis/response to therapy as detailed by the authors. So one could reasonably suggest that in breast cancer high TIL are actually in some ways associated with worse prognosis (if you look independently of type of BC) and authors should discuss this.

Line 94-The authors should amend the text as all mononuclear cells are not lymphocytes and plasma cells as monocytes are also mononuclear.

Lone 354. The authors should change slides should be "stained" shortly before IHC to "sectioned" before IHC.

There are minor errors in the English language such as certain words being missing due to English potentially not being the first language of the authors. This could be corrected at the editing stage by the journal as there aren't extensive errors and the English is still perfectly legible and acceptable.

Author Response

We thank the reviewer for its thoughtful comments and suggestions. We have incorporated changes that reflect the detailed suggestions you have graciously provided. To facilitate your review of our revisions, the following is a point-by-point response to the questions and comments. All changes in the main text have been highlighted in yellow.

Q1: The tables should be improved by adding references and a legend detailing whether/which regulatory body has made those recommendations (e.g. FDA).

A1: According to reviewer’s suggestion all tables have been modified with the inclusion of legends, references and detail on FDA recommendations.

Q2: The authors should discuss why the breast cancer subtype with the worst outcome (TNBC) has the highest number of TILs and yet they are prognostic and predictive of better response to chemotherapy.

A2: According to reviewer’s suggestion, prognostic and predictive role of TILs in TNBC (despite worst outcome) has been discussed in section 2.1 (lines 148-154).

Q3: Furthermore, in some of the other breast cancer subtypes high TILs have been shown in some studies to be associated with a worse prognosis/response to therapy as detailed by the authors. So one could reasonably suggest that in breast cancer high TIL are actually in some ways associated with worse prognosis (if you look independently of type of BC) and authors should discuss this.

A3: This interesting topic has been discussed in the conclusion section, as suggested.

Q4-Q5: Line 94-The authors should amend the text as all mononuclear cells are not lymphocytes and plasma cells as monocytes are also mononuclear. Line 354. The authors should change slides should be "stained" shortly before IHC to "sectioned" before IHC.

A4-A5: We have corrected.

Reviewer 3 Report

The increasing use of immunotherapies in breast cancer treatment has led to a growing body of research on the immunogenicity of the disease. Clinical studies have focused on identifying biomarkers that can reflect breast cancer's immunogenicity and predict response to immunotherapy. One such biomarker is tumor-infiltrating lymphocytes (TILs), which are more commonly found in triple-negative breast cancer and HER2+ subtypes. Higher levels of TILs have been associated with better response to neoadjuvant chemotherapy and improved survival. Another important biomarker is PD-L1, an inhibitory checkpoint protein involved in cancer immune regulation. PD-L1 expression in breast cancer varies based on tumor stage and molecular subtypes, with triple-negative breast cancer showing the highest rates of positivity. The prognostic role of PD-L1 expression is still controversial due to variations in immunohistochemistry methods and scoring systems used across studies. This paper provides a review of the immune landscape of breast cancer, discussing TILs and PD-L1 expression across different subtypes and offering guidance for their pathological assessment and reporting.

This paper has little novelty, no figures for illustration and no future outlook.

Author Response

We thank the reviewer for its comments and suggestion.

We believe that the growing body of research on the immunogenicity of breast cancer has generated confusion especially in the community of pathologists and oncologists which have to deal with TILs and PD-L1 assessment in daily practice. In this perspective our review aims to clarify and summarize this issues.

Despite illustrations are missing we hope the additional changes to improve manuscript quality will be appreciated.

Reviewer 4 Report

In this review, the authors have attempted to summarize the current practices for pathological evaluation of TILs and PD-L1 in breast cancer. The review topic is an interesting one but the body of the text is poorly organized with lack of clarity in message and focus. There are several areas of concern that must be addressed-

1. The title of the review is extremely misleading as neither the abstract nor the body of the text describe immune infiltrate in any amount of depth other than touching upon TIL analysis. It is very well appreciated that the breast tumor microenvironment is comprised of macrophages, MDSCs, NK cells and neutrophils in addition to T and B cells. The authors need to modify the title to TIL instead of tumor microenvironment or add details on infiltration by other immune cell types in breast tumors of all subtypes.

2. At the outset, the authors must provide some background and introduction to the different subsets of breast cancer. Of note, the review is missing any mention of lobular carcinoma within ER+ tumors, which has been shown to be a distinct histological subset with separate implications for immunotherapy.

3. None of the tables have any legends and are missing relevant references from where these data are curated. The authors claim to discuss TIL cut-offs in the summary but have not discussed this in the text. Either delete that line from the summary or add some discussion for TIL cut-off. It will be important to add those in Table 1

4. A brief introduction to PD-L1 pathway must be provided in order to orient the readers to the importance of this checkpoint in the clinic.

5. Table 3 must list all the relevant trials for each therapeutic category

6. It is unclear what section 3.4 is trying to achieve. The points described in that section appear to be some kind of protocol, which is vague enough to not be of any real value to someone trying to perform PD-L1 staining. This section can be deleted or in the least, references must be added that support the validity of the suggested steps.

7. Section 3.5 should be absorbed into section 3.3 

In general, the review will benefit from moderate reorganizing of sections and information contained within.

The authors have repeatedly used "In detail" incorrectly, where they probably need to use "Briefly". There are several typographical and grammatical errors that need to be addressed.

Author Response

We thank the reviewer for its thoughtful comments and suggestions. We have incorporated changes that reflect the detailed suggestions you have graciously provided. To facilitate your review of our revisions, the following is a point-by-point response to the questions and comments. All changes in the main text have been highlighted in yellow.

Q1. The title of the review is extremely misleading as neither the abstract nor the body of the text describe immune infiltrate in any amount of depth other than touching upon TIL analysis. It is very well appreciated that the breast tumor microenvironment is comprised of macrophages, MDSCs, NK cells and neutrophils in addition to T and B cells. The authors need to modify the title to TIL instead of tumor microenvironment or add details on infiltration by other immune cell types in breast tumors of all subtypes.

A1: We agree with the reviewer that title should be changed since “immune microenviroment” in BC is extremely heterogeneous and the focus of our research is the evaluation of TILs and PD-L1 which are routinely performed in breast pathology practice. We therefore utilized the term TILs, as suggested.

Q2. At the outset, the authors must provide some background and introduction to the different subsets of breast cancer. Of note, the review is missing any mention of lobular carcinoma within ER+ tumors, which has been shown to be a distinct histological subset with separate implications for imumunotherapy.

A2: According to reviewer’s suggestion, the introduction (lines 48-67) has been expanded to include additional insights on BC subtypes. Additionally, immunotherapy in ER+ lobular carcinoma as been discussed in section 3.2 (lines 322-335).

Q3. None of the tables have any legends and are missing relevant references from where these data are curated. The authors claim to discuss TIL cut-offs in the summary but have not discussed this in the text. Either delete that line from the summary or add some discussion for TIL cut-off. It will be important to add those in Table 1

A3: According to reviewer’s suggestion all tables have been modified with the inclusion of legends, references and TILs cut-offs (also discussed in lines 137-139).

Q4. A brief introduction to PD-L1 pathway must be provided in order to orient the readers to the importance of this checkpoint in the clinic.

A4: As suggested, we included in the introduction additional insights on PD-L1 pathway (lines 88-96).

Q5. Table 3 must list all the relevant trials for each therapeutic category

A5: all relevant trials have been included in the table, as suggested.

Q6. It is unclear what section 3.4 is trying to achieve. The points described in that section appear to be some kind of protocol, which is vague enough to not be of any real value to someone trying to perform PD-L1 staining. This section can be deleted or in the least, references must be added that support the validity of the suggested steps.

A6: after a careful revision of the manuscript, we agree with the reviewer that section 3.4 does not add any real value to our work so it was deleted.

Q7. Section 3.5 should be absorbed into section 3.3 

A7: We have done.

We also changed "in detail" with "briefly" as suggested

Round 2

Reviewer 4 Report

The authors have addressed all comments and made necessary changes. The review reads well!

Author Response

We thank the reviewer for its positive comments.